# Learning Differentiable Programs with Admissible Neural Heuristics

**Ameesh Shah**[*]

UC Berkeley

ameesh@berkeley.edu

**Eric Zhan**[*]

Caltech

ezhan@caltech.edu

**Jennifer J. Sun**

Caltech

jjsun@caltech.edu

**Abhinav Verma**

UT Austin

verma@utexas.edu

**Yisong Yue**

Caltech

yyue@caltech.edu

**Swarat Chaudhuri**

UT Austin

swarat@cs.utexas.edu

## Abstract

We study the problem of learning differentiable functions expressed as programs in a domain-specific language. Such programmatic models can offer benefits such as composability and interpretability; however, learning them requires optimizing over a combinatorial space of program "architectures". We frame this optimization problem as a search in a weighted graph whose paths encode top-down derivations of program syntax. Our key innovation is to view various classes of neural networks as continuous relaxations over the space of programs, which can then be used to complete any partial program. This relaxed program is differentiable and can be trained end-to-end, and the resulting training loss is an approximately admissible heuristic that can guide the combinatorial search. We instantiate our approach on top of the A* algorithm and an iteratively deepened branch-and-bound search, and use these algorithms to learn programmatic classifiers in three sequence classification tasks. Our experiments show that the algorithms outperform state-of-the-art methods for program learning, and that they discover programmatic classifiers that yield natural interpretations and achieve competitive accuracy.

## 1 Introduction

An emerging body of work advocates *program synthesis* as an approach to machine learning. The methods here learn functions represented as programs in symbolic, domain-specific languages (DSLs) [12, 11, 49, 44, 46, 45]. Such symbolic models have a number of appeals: they can be more interpretable than neural models, they use the inductive bias embodied in the DSL to learn reliably, and they use compositional language primitives to transfer knowledge across tasks.

In this paper, we study how to learn *differentiable* programs, which use structured, symbolic primitives to compose a set of parameterized, differentiable modules. Differentiable programs have recently attracted much interest due to their ability to leverage the complementary advantages of programming language abstractions and differentiable learning. For example, recent work has used such programs to compactly describe modular neural networks that operate over rich, recursive data types [44].

To learn a differentiable program, one needs to induce the program's "architecture" while simultaneously optimizing the parameters of the program's modules. This co-design task is difficult because the space of architectures is combinatorial and explodes rapidly. Prior work has approached this challenge using methods such as greedy enumeration, Monte Carlo sampling, Monte Carlo tree

---

[*]Equal Contribution.

search, and evolutionary algorithms [46, 44, 10]. However, such approaches can often be expensive, due to not fully exploiting the structure of the underlying combinatorial search problem.

In this paper, we show that the differentiability of programs opens up a new line of attack on this search problem. A standard strategy for combinatorial optimization is to exploit (ideally fairly tight) continuous relaxations of the search space [31, 6, 48, 42, 24, 2, 45]. Optimization in the relaxed space is typically easier and can efficiently guide search algorithms towards good or optimal solutions. In the case of program learning, we propose to use various classes of neural networks as relaxations of partial programs. We frame our problem as searching a graph, in which nodes encode program architectures with missing expressions, and paths encode top-down program derivations. For each partial architecture $u$ encountered during this search, the relaxation amounts to substituting the unknown part of $u$ with a neural network with free parameters. Because programs are differentiable, this network can be trained on the problem's end-to-end loss. If the space of neural networks is an (approximate) proper relaxation of the space of programs (and training identifies a near-optimum neural network), then the training loss for the relaxation can be viewed as an (approximately) admissible heuristic.

We instantiate our approach, called NEAR (abbreviation for *Neural Admissible Relaxation*), on top of two informed search algorithms: A* and an iteratively deepened depth-first search that uses a heuristic to direct branching as well as branch-and-bound pruning (IDS-BB). We evaluate the algorithms in the task of learning programmatic classifiers in three behavior classification applications. We show that the algorithms substantially outperform state-of-the-art methods for program learning, and can learn classifier programs that bear natural interpretations and are close to neural models in accuracy.

To summarize, the paper makes three contributions. First, we identify a tool — heuristics obtained by training neural relaxations of programs — for accelerating combinatorial searches over differentiable programs. So far as we know, this is the first approach to exploit the differentiability of a programming language in program synthesis. Second, we instantiate this idea using two classic search algorithms. Third, we present promising experimental results in three sequence classification applications.

## 2 Problem Formulation

We view a program in our domain-specific language (DSL) as a pair $(\alpha, \theta)$, where $\alpha$ is a discrete *(program) architecture* and $\theta$ is a vector of real-valued parameters. The architecture $\alpha$ is generated using a *context-free grammar* [21]. The grammar consists of a set of rules $X \to \sigma_1 \ldots \sigma_k$, where $X$ is a *nonterminal* and $\sigma_1, \ldots, \sigma_k$ are either nonterminals or *terminals*. A nonterminal stands for a missing subexpression; a terminal is a symbol that can actually appear in a program's code. The grammar starts with an initial nonterminal, then iteratively applies the rules to produce a series of *partial architectures*: sentences made from one or more nonterminals and zero or more terminals. The process continues until there are no nonterminals left, i.e., we have a complete architecture.

The *semantics* of the architecture $\alpha$ is given by a function $[\![\alpha]\!](x, \theta)$, defined by rules that are fixed for the DSL. We require this function to be differentiable in $\theta$. Also, we define a *structural cost* for architectures. Let each rule $r$ in the DSL grammar have a non-negative real cost $s(r)$. The structural cost of $\alpha$ is $s(\alpha) = \sum_{r \in \mathcal{R}(\alpha)} s(r)$, where $\mathcal{R}(\alpha)$ is the multiset of rules used to create $\alpha$. Intuitively, architectures with lower structural cost are simpler are more human-interpretable.

To define our learning problem, we assume an unknown distribution $D(x, y)$ over inputs $x$ and labels $y$, and consider the prediction error function $\zeta(\alpha, \theta) = \mathbb{E}_{(x,y) \sim D}[\mathbf{1}([\![\alpha]\!](x, \theta) \neq y)]$, where $\mathbf{1}$ is the indicator function. Our goal is to find an architecturally simple program with low prediction error, i.e., to solve the optimization problem:

$$(\alpha^*, \theta^*) = \underset{(\alpha, \theta)}{\arg \min}(s(\alpha) + \zeta(\alpha, \theta)). \tag{1}$$

**Program Learning for Sequence Classification.** Program learning is applicable in many settings; we specifically study it in the sequence classification context [9]. Now we sketch our DSL for this domain. Like many others DSLs for program synthesis [15, 3, 44], our DSL is purely functional. The language has the following characteristics:

- Programs in the DSL operate over two data types: real vectors and sequences of real vectors. We assume a simple type system that makes sure that these types are used consistently.

$$\alpha \quad ::= \quad x \mid c \mid \oplus(\alpha_1, \ldots, \alpha_k) \mid \oplus_\theta(\alpha_1, \ldots, \alpha_k) \mid \textbf{if } \alpha_1 \textbf{ then } \alpha_2 \textbf{ else } \alpha_3 \mid \textbf{sel}_S \, x$$
$$\textbf{map } (\lambda x_1.\alpha_1) \, x \mid \textbf{fold } (\lambda x_1.\alpha_1) \, c \, x \mid \textbf{mapprefix } (\lambda x_1.\alpha_1) \, x$$

Figure 1: Grammar of DSL for sequence classification. Here, $x$, $c$, $\oplus$, and $\oplus_\theta$ represent inputs, constants, basic algebraic operations, and parameterized library functions, respectively. $\textbf{sel}_S$ returns a vector consisting of a subset $S$ of the dimensions of an input $x$.

$$\textbf{map}(\textbf{if } DistAffine_{[.0217];-.2785}(x)$$
$$\textbf{then } AccAffine_{[-.0007,.0055,.0051,-.0025];3.7426}(x) \textbf{ else } DistAffine_{[-.2143];1.822}(x)$$

Figure 2: Synthesized program classifying a "sniff" action between two mice in the CRIM13 dataset. $DistAffine$ and $AccAffine$ are functions that first select the parts of the input that represent distance and acceleration measurements, respectively, and then apply affine transformations to the resulting vectors. In the parameters (subscripts) of these functions, the brackets contain the weight vectors for the affine transformation, and the succeeding values are the biases. The program achieves an accuracy of 0.87 (vs. 0.89 for RNN baseline) and can be interpreted as follows: if the distance between two mice is small, they are doing a "sniff" (large bias in **else** clause). Otherwise, they are doing a "sniff" if the difference between their accelerations is small.

- Programs use a set of fixed algebraic operations $\oplus$ as well as a "library" of differentiable, parameterized functions $\oplus_\theta$. Because we are motivated by interpretability, the library used in our current implementation only contains affine transformations. In principle, it could be extended to include other kinds of functions as well.

- Programs use a set of higher-order combinators to recurse over sequences. In particular, we allow the standard **map** and **fold** combinators. To compactly express sequence-to-sequence functions, we also allow a special **mapprefix** combinator. Let $g$ be a function that maps sequences to vectors. For a sequence $x$, $\textbf{mapprefix}(g, x)$ equals the sequence $\langle g(x_{[1:1]}), g(x_{[1:2]}), \ldots, g(x_{[1:n]}) \rangle$, where $x_{[1:i]}$ is the $i$-th prefix of $x$.

- Programs can use a conditional branching construct. However, to avoid discontinuities, we interpret this construct in terms of a smooth approximation:
$$[\![\textbf{if } \alpha_1 > 0 \textbf{ then } \alpha_2 \textbf{ else } \alpha_3]\!](x, (\theta_1, \theta_2, \theta_3))$$
$$= \sigma(\beta \cdot [\![\alpha_1]\!](x, \theta_1)) \cdot [\![\alpha_2]\!](x, \theta_2) + (1 - \sigma(\beta \cdot [\![\alpha_1]\!](x, \theta_1))) \cdot [\![\alpha_3]\!](x, \theta_3). \tag{2}$$
Here, $\sigma$ is the sigmoid function and $\beta$ is a temperature hyperparameter. As $\beta \to 0$, this approximation approaches the usual if-then-else construct.

Figure 1 summarizes our DSL in the standard Backus-Naur form [47]. Figures 2 and 3 show two programs synthesized by our learning procedure using our DSL with libraries of domain-specific affine transformations (see the supplementary material). Both programs offer an interpretation in their respective domains, while offering respectable performance against an RNN baseline.

## 3 Program Learning using NEAR

We formulate our program learning problem as a form of graph search. The search derives program architectures top-down: it begins with the *empty* architecture, generates a series of partial architectures following the DSL grammar, and terminates when a complete architecture is derived.

In more detail, we imagine a graph $\mathcal{G}$ in which:

- The node set consists of all partial and complete architectures permissible in the DSL.

- The *source node* $u_0$ is the empty architecture. Each complete architecture $\alpha$ is a *goal node*.

- Edges are directed and capture single-step applications of rules of the DSL. Edges can be divided into: (i) *internal edges* $(u, u')$ between partial architectures $u$ and $u'$, and (ii) *goal edges* $(u, \alpha)$ between partial architecture $u$ and complete architecture $\alpha$. An internal edge $(u, u')$ exists if one can obtain $u'$ by substituting a nonterminal in $u$ following a rule of the DSL. A goal edge $(u, \alpha)$ exists if we can complete $u$ into $\alpha$ by applying a rule of the DSL.

$$\textbf{map}(\textbf{multiply}(\textbf{add}(\textit{OffenseAffine}(x), \textit{BallAffine}(x)), \textbf{add}(\textit{OffenseAffine}(x), \textit{BallAffine}(x)))$$

Figure 3: Synthesized program classifying the ballhandler for basketball. *OffenseAffine()* and *BallAffine()* are parameterized affine transformations over the XY-coordinates of the offensive players and the ball (see the appendix for full parameters). **multiply** and **add** are computed element-wise. The program structure can be interpreted as computing the Euclidean norm/distance between the offensive players and the ball and suggests that this quantity can be important for determining the ballhandler. On a set of learned parameters (not shown), this program achieves an accuracy of 0.905 (vs. 0.945 for an RNN baseline).

- The cost of an internal edge $(u, u')$ is given by the structural cost $s(r)$, where $r$ is the rule used to construct $u'$ from $u$. The cost of a goal edge $(u, \alpha)$ is $s(r) + \zeta(\alpha, \theta^*)$, where $\theta^* = \arg\min_\theta \zeta(\alpha, \theta)$ and $r$ is the rule used to construct $\alpha$ from $u$.

A path in the graph $\mathcal{G}$ is defined as usual, as a sequence of nodes $u_1, \dots, u_k$ such that there is an edge $(u_i, u_{i+1})$ for each $i \in \{1, \dots, k-1\}$. The cost of a path is the sum of the costs of these edges. Our goal is to discover a least-cost path from the source $u_0$ to some goal node $\alpha^*$. Then by construction of our edge costs, $\alpha^*$ is an optimal solution to our learning problem in Eq. (1).

## 3.1 Neural Relaxations as Admissible Heuristics

The main challenge in our search problem is that our goal edges contain rich cost information, but this information is only accessible when a path has been explored until the end. A heuristic function $h(u)$ that can predict the value of choices made at nodes $u$ encountered early in the search can help with this difficulty. If such a heuristic is *admissible* — i.e., underestimates the cost-to-go — it enables the use of informed search strategies such as A$^*$ and branch-and-bound while guaranteeing optimal solutions. Our NEAR approach (abbreviation for *Neural Admissible Relaxation*) uses neural approximations of spaces of programs to construct a heuristic that is $\epsilon$-close to being admissible.

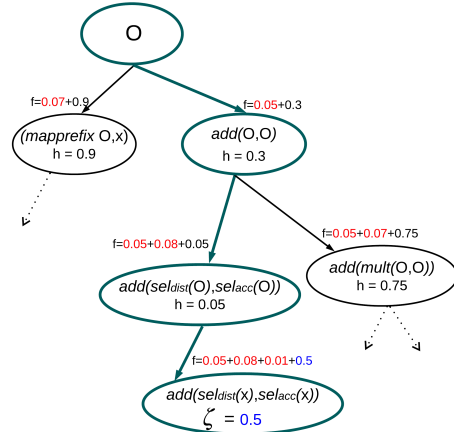

Figure 4: An example of program learning formulated as graph search. Structural costs are in red, heuristic values in black, prediction errors $\zeta$ in blue, O refers to a nonterminal in a partial architecture, and the path to a goal node returned by A*-NEAR search is in teal.

Let a *completion* of a partial architecture $u$ be a (complete) architecture $u[\alpha_1, \dots, \alpha_k]$ obtained by replacing the nonterminals in $u$ by suitably typed architectures $\alpha_i$. Let $\theta_u$ be the parameters of $u$ and $\theta$ be parameters of the $\alpha_i$-s. The *cost-to-go* at $u$ is given by:

$$J(u) = \min_{\alpha_1, \dots, \alpha_k, \theta_u, \theta} ((s(u[\alpha_1, \dots, \alpha_k] - s(u)) + \zeta(u[\alpha_1, \dots, \alpha_k], (\theta_u, \theta)) \qquad (3)$$

where the structural cost $s(u)$ is the sum of the costs of the grammatical rules used to construct $u$.

To compute a heuristic cost $h(u)$ for a partial architecture $u$ encountered during search, we substitute the nonterminals in $u$ with neural networks parameterized by $\omega$. These networks are *type-correct* — for example, if a nonterminal is supposed to generate subexpressions whose inputs are sequences, then the neural network used in its place is recurrent. We show an example of NEAR used in a program learning-graph search formulation in Figure 4.

We view the neurosymbolic programs resulting from this substitution as tuples $(u, (\theta_u, \omega))$. We define a semantics for such programs by extending our DSL's semantics, and lift the function $\zeta$ to assign costs $\zeta(u, (\theta_u, \omega))$ to such programs. The heuristic cost for $u$ is now given by:

$$h(u) = \min_{w, \theta} \zeta(u, (\theta_u, \omega)). \qquad (4)$$

As $\zeta(u, (\theta_u, \omega))$ is differentiable in $\omega$ and $\theta_u$, we can compute $h(u)$ using gradient descent.

$\epsilon$-**Admissibility.** In practice, the neural networks that we use may only form an approximate relaxation of the space of completions and parameters of architectures; also, the training of these networks may not reach global optima. To account for these errors, we consider an approximate notion of admissibility. Many such notions have been considered in the past [20, 31, 43]; here, we follow a definition used by Harris [20]. For a fixed constant $\epsilon > 0$, let an $\epsilon$-*admissible heuristic* be a function $h^*(u)$ over architectures such that $h^*(u) \leq J(u) + \epsilon$ for all $u$. Now consider any completion $u[\alpha_1, \ldots, \alpha_k]$ of an architecture $u$. As neural networks with adequate capacity are universal function approximators, there exist parameters $\omega^*$ for our neurosymbolic program such that for all $u, \alpha_1, \ldots, \alpha_k, \theta_u$, and $\theta$:

$$\zeta(u, (\theta_u, \omega^*)) \leq \zeta(u[\alpha_1, \ldots, \alpha_k], (\theta_u, \theta)) + \epsilon. \tag{5}$$

Because edges in our search graph have non-negative costs, $s(u) \leq s(u[\alpha_1, \ldots, \alpha_k])$, implying:

$$
\begin{aligned}
h(u) &\leq \min_{\alpha_1, \ldots, \alpha_k, \theta_u, \theta} \zeta(u[\alpha_1, \ldots, \alpha_k], (\theta_u, \theta)) + \epsilon \\
&\leq \min_{\alpha_1, \ldots, \alpha_k, \theta_u, \theta} \zeta(u[\alpha_1, \ldots, \alpha_k], (\theta_u, \theta)) + (s(u[\alpha_1, \ldots, \alpha_k]) - s(u)) + \epsilon = J(u) + \epsilon.
\end{aligned}
\tag{6}
$$

In other words, $h(u)$ is $\epsilon$-admissible.

**Empirical Considerations.** We have formulated our learning problem in terms of the true prediction error $\zeta(\alpha, \theta)$. In practice, we must use statistical estimates of this error. Following standard practice, we use an empirical validation error to choose architectures, and an empirical training error is used to choose module parameters. This means that in practice, the cost of a goal edge $(u, \alpha)$ in our graph is $\zeta^{val}(\alpha, \arg\min_\theta \zeta^{train}(\alpha, \theta))$.

One complication here is that our neural heuristics encode both the completions of an architecture and the parameters of these completions. Training a heuristic on either the training loss or the validation loss will introduce an additional error. Using standard generalization bounds, we can argue that for adequately large training and validation sets, this error is bounded (with probability arbitrarily close to 1) in either case, and that our heuristic is $\epsilon$-admissible with high probability in spite of this error.

## 3.2 Integrating NEAR with Graph Search Algorithms

The NEAR approach can be used in conjunction with any heuristic search algorithm [36] over architectures. Specifically, we have integrated NEAR with two classic graph search algorithms: $A^*$ [31] (Algorithm 1) and an iteratively deepened depth-first search with branch-and-bound pruning (IDS-BB) (Appendix A). Both algorithms maintain a *search frontier* by computing an *f-score* for each node: $f(u) = g(u) + h(u)$, where $g(u)$ is the incurred path cost from the source node $u_0$ to the current node $u$, and $h(u)$ is a heuristic estimate of the cost-to-go from node $u$. Additionally, IDS-BB prunes nodes from the frontier that have a higher $f$-score than the minimum path cost to a goal node found so far.

---

**Algorithm 1:** A* Search

**Input:** Graph $\mathcal{G}$ with source $u_0$
$S := \{u_0\}; f(u_0) := \infty;$
**while** $S \neq \emptyset$ **do**
  $v := \arg\min_{u \in S} f(u);$
  $S := S \setminus \{v\};$
  **if** *v is a goal node* **then**
    | **return** $v, f_v;$
  **else**
    | **foreach** *child u of v* **do**
    |   Compute $g(u), h(u), f(u);$
    |   $S := S \cup \{u\};$

---

$\epsilon$-**Optimality.** An important property of a search algorithm is *optimality*: when multiple solutions exist, the algorithm finds an optimal solution. Both $A^*$ and IDS-BB are optimal given admissible heuristics. An argument by Harris [20] shows that under heuristics that are $\epsilon$-admissible in our sense, the algorithms return solutions that at most an additive constant $\epsilon$ away from the optimal solution. Let $C^*$ denote the optimal path cost in our graph $\mathcal{G}$, and let $h(u)$ be an $\epsilon$-admissible heuristic (Eq. (6)). Suppose IDS-BB or $A^*$ returns a goal node $\alpha_G$ that does not have the optimal path cost $C^*$. Then there must exist a node $u_O$ on the frontier that lies along the optimal path and has yet to be expanded. This lets us establish an upper bound on the path cost of $\alpha_G$:

$$g(\alpha_G) = f(\alpha_G) \leq f(u_O) = g(u_O) + h(u_O) \leq g(u_O) + J(u_O) + \epsilon \leq C^* + \epsilon. \tag{7}$$

This line of reasoning can also be extended to the Branch-and-Bound component of the NEAR-guided IDS-BB algorithm. Consider encountering a goal node during search that sets the branch-and-bound upper threshold to be a cost $C$. In the remainder of search, some node $u_p$ with an $f$-cost greater than $C$ is pruned, and the optimal path from $u_p$ to a goal node will not be searched. Assuming the heuristic function $h$ is $\epsilon$-admissible, we can set a lower bound on the optimal path cost from $u_p$, $f(u_p^*)$, to be $C - \epsilon$ by the following:

$$f(u_p^*) = g(u_p) + J(u_p) \geq f(u_p) = g(u_p) + h(u_p) + \epsilon > C = g(u_p) + h(u_p) > C - \epsilon. \quad (8)$$

Thus, the IDS-BB algorithm will find goal paths are at worst an additive factor of $\epsilon$ more than any pruned goal path.

## 4 Experiments

### 4.1 Datasets for Sequence Classification

For all datasets below, we augment the base DSL in Figure 1 with domain-specific library functions that include 1) learned affine transformations over a subset of features, and 2) sliding window feature-averaging functions. Full details, such as structural cost functions used and any pre/post-processing, are provided in the appendix.

**CRIM13.** The *CRIM13* dataset [5] contains trajectories for a pair of mice engaging in social behaviors, annotated for different actions per frame by behavior experts; we aim to learn programs for classifying actions at each frame for fixed-size trajectories. Each frame is represented by a 19-dimensional feature vector: 4 features capture the $xy$-positions of the mice, and the remaining 15 features are derived from the positions, such as velocities and distance between mice. We learn programs for two actions that can be identified the tracking features: "sniff" and "other" ("other" is used when there is no behavior of interest occurring). We cut every 100 frames as a trajectory, and in total we have 12404 training, 3077 validation, and 2953 test trajectories.

**Fly-vs.-Fly.** We use the *Aggression* and *Boy-meets-Boy* datasets within the *Fly-vs.-Fly* environment that tracks a pair of fruit flies and their actions as they interact in different contexts [14]. We aim to learn programs that classify trajectories as one of 7 possible actions displaying aggressive, threatening, and nonthreatening behaviors. The length of trajectories can range from 1 to over 10000 frames, but we segment the data into trajectories with a maximum length of 300 for computational efficiency. The average length of a trajectory in our training set is 42.06 frames. We have 5339 training, 594 validation, and 1048 test trajectories.

**Basketball.** We use a subset of the basketball dataset from [50] that tracks the movements of professional basketball players. Each trajectory is of length 25 and contains the $xy$-positions of 5 offensive players, 5 defensive players, and the ball (22 features per frame). We aim to learn programs that can predict which offensive player has the ball (the "ballhandler") or whether the ball is being passed. In total, we have 18,000 trajectories for training, 2801 for validation, and 2693 for test.

### 4.2 Overview of Baseline Program Learning Strategies

We compare our NEAR-guided graph search algorithms, A*-NEAR and IDS-BB-NEAR, with four baseline program learning strategies: 1) top-down enumeration, 2) Monte-Carlo sampling, 3) Monte-Carlo tree search, and 4) a genetic algorithm. We also compare the performance of these program learning algorithms with an RNN baseline (1-layer LSTM).

**Top-down enumeration.** We synthesize and evaluate complete programs in order of increasing complexity measured using the structural cost $s(\alpha)$. This strategy is widely employed in program learning contexts [44, 46, 45] and is provably complete. Since our graph $\mathcal{G}$ grows infinitely, our implementation is akin to breadth-first search up to a specified depth.

**Monte-Carlo (MC) sampling.** Starting from the source node $u_0$, we sample complete programs by sampling rules (edges) with probabilities proportional to their structural costs $s(r)$. The next node chosen along a path has the best average performance of samples that descended from that node. We repeat the procedure until we reach a goal node and return the best program found among all samples.

**Monte-Carlo tree search (MCTS).** Starting from the source node $u_0$, we traverse the graph until we reach a complete program using the UCT selection criteria [23], where the value of a node is inversely proportional to the cost of its children.[2] In the backpropagation step we update the value of all nodes along the path. After some iterations, we choose the next node in the path with the highest value. We repeat the procedure until we reach a goal node and return the best program found.

|  | **CRIM13-sniff** | | | **CRIM13-other** | | | **Fly-vs.-Fly** | | | **Bball-ballhandler** | | |
|---|---|---|---|---|---|---|---|---|---|---|---|---|
|  | Acc. | F1 | Depth | Acc. | F1 | Depth | Acc. | F1 | Depth | Acc. | F1 | Depth |
| Enum. | .851 | .221 | 3 | .707 | .762 | 2 | .819 | .863 | 2 | .844 | .857 | 6.3 |
| MC | .843 | .281 | 7 | .630 | .715 | 1 | .833 | .852 | 4 | .841 | .853 | 6 |
| MCTS | .745 | .338 | 8.7 | .666 | .749 | 1 | .817 | .857 | 4.7 | .711 | .729 | 8 |
| Genetic | .829 | .181 | 1.7 | .727 | .768 | 3 | .850 | .868 | 6 | .843 | .853 | 6.7 |
| IDS-BB-NEAR | .829 | .446 | 6 | .729 | .768 | 1.3 | .876 | .892 | 4 | .889 | .903 | 8 |
| A*-NEAR | .821 | .369 | 6 | .706 | .764 | 2.7 | .872 | .885 | 4 | .906 | .918 | 8 |
| RNN | .889 | .481 | - | .756 | .785 | - | .963 | .964 | - | .945 | .950 | - |

Table 1: Mean accuracy, F1-score, and program depth of learned programs (3 trials). Programs found using our NEAR algorithms consistently achieve better F1-score than baselines and match more closely to the RNN's performance. Our algorithms are also able to search and find programs of much greater depth than the baselines. Experiment hyperparameters are included in the appendix.

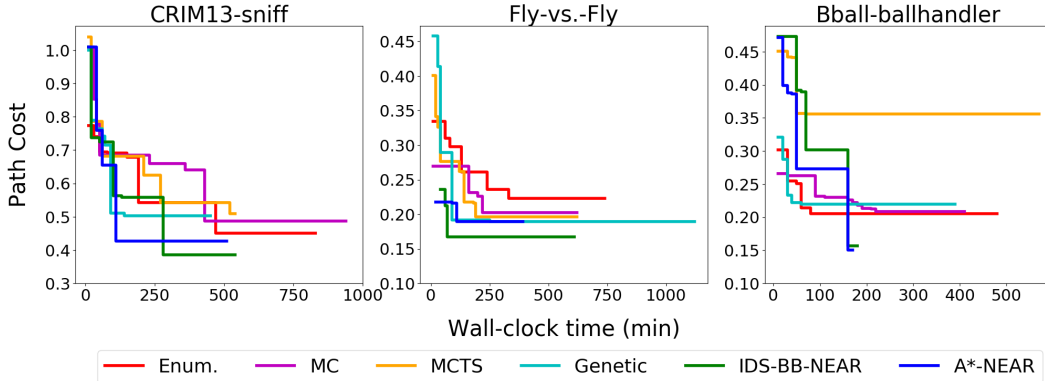

Figure 5: Median minimum path cost to a goal node found at a given time, across 3 trials (for trials that terminate first, we extend the plots so the median remains monotonic). A*-NEAR (blue) and IDS-BB-NEAR (green) will often find a goal node with a smaller path cost, or find one of similar performance but much faster.

**Genetic algorithm.** We follow the formulation in Valkov et al. [44]. In our genetic algorithm, crossover, selection, and mutation operations evolve a population of programs over a number of generations until a predetermined number of programs have been trained. The crossover and mutation operations only occur when the resulting program is guaranteed to be type-safe.

For all baseline algorithms, as well as A*-NEAR and IDS-BB-NEAR, model parameters ($\theta$) were learned with the training set, whereas program architectures ($\alpha$) were evaluated using the performance on the validation set. Additionally, all baselines (including NEAR algorithms) used F1-score [38] error as the evaluation objective $\zeta$ by which programs were chosen. To account for class imbalances, F1-scoring is commonly used as an evaluation metric in behavioral classification domains, such as those considered in our work [14, 5]. Our full implementation is available in [39].

### 4.3 Experimental Results

**Performance of learned programs.** Table 1 shows the performance results on the test sets of our program learning algorithms, averaged over 3 seeds. The same structural cost function $s(\alpha)$ is used for all algorithms, but can vary across domains (see Appendix). Our NEAR-guided search algorithms consistently outperform other baselines in F1-score while accuracy is comparable (note that our $\zeta$ does not include accuracy). Furthermore, NEAR-guided search algorithms are capable are finding deeper and more complex programs that can offer non-trivial interpretations, such as the ones shown in Figures 2 and 3. Lastly, we verify that our learned programs are comparable with highly expressive RNNs, and see that there is at most a 10% drop in F1-score when using NEAR-guided search algorithms with our DSL.

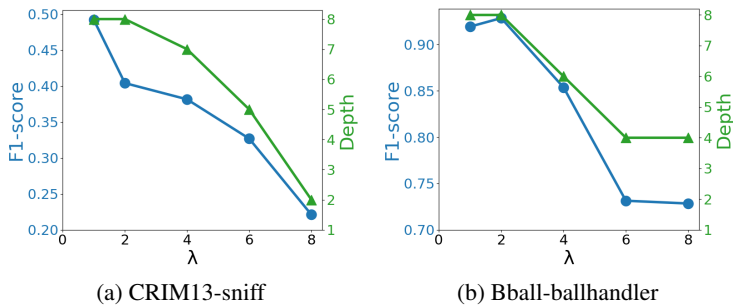

(a) CRIM13-sniff       (b) Bball-ballhandler

Figure 6: As we increase $\lambda$ in Eq. (9), we observe that A*-NEAR will learn programs with decreasing program depth and also decreasing F1-score. This highlights that we can use $\lambda$ to control the trade-off between structural cost and performance.

---

**mapprefix(SlidingWindowAverage(** *PositionAffine(x)* **)))**

Figure 7: Synthesized depth 2 program classifying a "sniff" action between two mice in the CRIM13 dataset. The sliding window average is over the last 10 frames. The program achieves F1 score of 0.22 (vs. 0.48 for RNN baseline). This program is synthesized using $\lambda = 8$.

---

**Efficiency of NEAR-guided graph search.** Figure 5 tracks the progress of each program learning algorithm during search by following the median best path cost (Eq. (1)) at a given time across 3 independent trials. For times where only 2 trials are active (i.e. one trial had already terminated), we report the average. Algorithms for each domain were run on the same machine to ensure consistency, and each non-NEAR baseline was set up such to have at least as much time as our NEAR-guided algorithms for their search procedures (see Appendix). We observe that NEAR-guided search algorithms are able to find low-cost solutions more efficiently than existing baselines, while maintaining an overall shorter running time.

**Cost-performance trade-off.** We can also consider a modification of our objective in Eq. (1) that allows us to use a hyperparameter $\lambda$ to control the trade-off between structural cost (a proxy for interpretability) and performance:

$$(\alpha^*, \theta^*) = \underset{(\alpha,\theta)}{\arg\min}(\lambda \cdot s(\alpha) + \zeta(\alpha, \theta)). \tag{9}$$

To visualize this trade-off, we run A*-NEAR with the modified objective Eq. (9) for various values of $\lambda$. Note that $\lambda = 1$ is equivalent to our experiments in Table 1. Figure 6 shows that for the *Basketball* and *CRIM13* datasets, as we increase $\lambda$, which puts more weight on the structural cost, the resulting programs found by A*-NEAR search have decreasing F1-scores but are also more shallow. This confirms our expectations that we can control the trade-off between structural cost and performance, which allows users of NEAR-guided search algorithms to adjust to their preferences. Unlike the other two experimental domains, the most performant programs learned in *Fly-vs.-Fly* were relatively shallow, so we omitted this domain as the trade-off showed little change in program depth.

We illustrate the implications of this tradeoff on interpretability using the depth-2 program in Figure 7 and the depth-8 program in Figure 8, both synthesized for the same task of detecting a "sniff" action in the CRIM13 dataset. The depth-2 program says that a "sniff" occurs if the intruder mouse is close to the right side of the cage and both mice are near the bottom of the cage, and can be seen to apply a *position bias* (regarding the location of the action) on the action. This program is simple, due to the large weight on the structural cost, and has a low F1-score. In contrast, the deeper program in Figure 8 has performance comparable to an RNN but is more difficult to interpret. Our interpretation of this program is that it evaluates the likelihood of "sniff" by applying a position bias, then using the velocity of the mice if the mice are close together and not moving fast, and using distance between the mice otherwise.

## 5 Related Work

**Neural Program Induction.** The literature on *neural program induction* (NPI) [19, 34, 25, 37] develops methods to learn neural networks that can perform procedural (program-like) tasks, typically using architectures augmented with differentiable memory. Our approach differs from these methods in that its final output is a symbolic program. However, since our heuristics are neural approximation

> **map**(**add**( *PositionAffine*(x),
>
>          **if add**( *VelocityAffine*(x), *DistAffine*(x))
>
>          **then** *VelocityAffine*(x) **else** *DistAffine*(x)))
>
> Figure 8: Synthesized depth 8 program classifying a "sniff" action between two mice in the CRIM13 dataset. The program achieves F1 score of 0.46 (vs. 0.48 for RNN baseline). This program is synthesized using $\lambda = 1$.

of programs, our work can be seen as repeatedly performing NPI as the program is being produced. While we have so far used classical feedforward and recurrent architectures to implement our neural heuristics, future work could use richer models from the NPI literature to this end.

**DSL-based Program Synthesis.** There is a large body of research on synthesis of programs from DSLs. In many of these methods, the goal is not learning but finding a program that satisfies a hard constraint [1, 41, 32, 15]. However, there is also a growing literature on learning programs from (noisy) data [26, 13, 46, 11, 44, 45]. Of these methods, TERPRET [18] and NEURAL TERPRET [17] allows gradient descent as a mechanism for learning program parameters. However, unlike NEAR, these approaches do not allow a general search over program architectures permitted by a DSL, and require a detailed hand-written template of the program for even the simplest tasks. While the Houdini framework [44] combines gradient-based parameter learning with search over program architectures, this search is not learning-accelerated and uses a simple type-directed enumeration. As reported in our experiments, NEAR outperforms this enumeration-based approach.

Many recent methods for program synthesis use statistical models to guide the search over program architectures [3, 7, 12, 8, 11, 30, 16, 29]. In particular, Lee et al. [27] use a probabilistic model to guide an A$^*$ search over programs. Most of these models (including the one in Lee et al. [27]) are trained using corpora of synthesis problems and corresponding solutions, which are not available in our setting. There is a category of methods based on reinforcement learning (RL) [16, 4]. Unlike NEAR, these methods do not directly exploit the structure of the search space. Combining them with our approach would be an interesting topic of future work.

**Structure Search using Relaxations.** Our search problem bears similarities with the problems of searching over neural architectures and the structure of graphical models. Prior work has used relaxations to solve these problems [28, 40, 51, 33, 48]. Specifically, the A* lasso approach for learning sparse Bayesian networks [48] uses a dense network to construct admissible heuristics, and DARTS computes a differentiable relaxation of neural architecture search [28, 40]. The key difference between these efforts and ours is that the design space in our problem is much richer, making the methods in prior work difficult to apply. In particular, DARTS uses a composition of softmaxes over all possible candidate operations between a fixed set of nodes that constitute a neural architecture, and the heuristics in the A* lasso method come from a single, simple function class. However, in our setting, there is no fixed bound on the number of expressions in a program, different sets of operations can be available at different points of synthesis, and the input and output type of the heuristic (and therefore, its architecture) can vary based on the part of the program derived so far.

## 6   Conclusions

We have a presented a novel graph search approach to learning differentiable programs. Our method leverages a novel construction of an admissible heuristic using neural relaxations to efficiently search over program architectures. Our experiments show that programs learned using our approach can have competitive performance, and that our search-based learning procedure substantially outperforms conventional program learning approaches.

There are many directions for future work. One direction is to extend the approach to richer DSLs and neural heuristic architectures, for example, those suited to reinforcement learning [45] and generative modeling [35]. Another is to combine NEAR with classical program synthesis methods based on symbolic reasoning. A third is to integrate NEAR into more complex program search problems, e.g., when there is an initial program as a starting point and the goal is to search for refinements. A fourth is to more tightly integrate with real-world applications to evaluate the interpretability of learned programs as it impacts downstream tasks.

## Broader Impact

Programmatic models described using high-level DSLs are a powerful mechanism for summarizing automatically discovered knowledge in a human-interpretable way. Specifically, such models are more interpretable than state-of-the-art neural models while also tending to provide higher performance than shallower linear or decision tree models. Also, programmatic models allow for natural incorporation of inductive bias and allow the user to influence the semantic meaning of learned programs.

For these reasons, efforts on program learning, such as ours, can lead to more widespread use of machine learning in fields, such as healthcare, autonomous driving, and the natural sciences, where safety and accountability are critical and there human-held prior knowledge (such as the laws of nature) that can usefully direct the learning process. The flipside of this is that the bias introduced in program learning can just as easily be exploited by users who desire specific outcomes from the learner. Ultimately, users of program learning methods must ensure that any incorporated inductive bias will not lead to unfair or misleading programs.

## Funding Acknowledgment

This work was supported in part by NSF Award # CCF-1918651, NSF Award # CCF-1704883, DARPA PAI, Raytheon, a Rice University Graduate Research Fellowship (for Shah), a JP Morgan Chase Fellowship (for Verma), and NSERC Award # PGSD3-532647-2019 (for Sun).

## Footnotes

[2]MCTS with this node value definition will visit shallow programs more frequently than MC sampling.

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
