[Supplementary Material · near_appendix.pdf]

## A  IDS-BB

In Algorithm 2, we provide the pseudocode for the IDS-BB algorithm introduced in the main text. This algorithm is a Heuristic-Guided Depth-First Search with three key characteristics: (1) the search depth is iteratively increased; (2) the search is ordered using a function $f(u)$ as in A$^*$, and (3) Branch-and-Bound is used to prune unprofitable parts of the search space. We find that the use of iterative deepening in the program learning setting is useful in that it prioritizes searching shallower and less parsimonious programs early on in the search process.

---

**Algorithm 2:** Iterative Deepening Depth-First-Search

---

**Input:** Initial depth $d_{initial}$, Max depth $d_{max}$
Initialize *frontier* to a priority-queue with root node *root* ;
Initialize *nextfrontier* to an empty priority-queue;
$(f_{root}, f_{min}, d_{iter}) = (\infty, \infty, d_{initial})$;
$current = None$;
**while** *frontier is not empty* **do**
    **if** *current is None* **then**
        pop node with lowest $f$ from *frontier* and assign to *current*;
    **if** *current is a leaf node* **then**
        $f_{min} := \min(f_{current}, f_{min})$;
        $current := None$;
    **else**
        **if** $d_{current} > d_{iter}$ **then**
            $current := None$;
        **else**
            Set *current* to child with lowest $f$;
            **if** $d_{current} \leq d_{max}$ **then**
                Evaluate and add all children of *current* to *frontier*;
            **if** *frontier is empty* **then**
                $frontier := nextfrontier$;
                $d_{iter} := d_{iter} + 1$;
**return** $f_{min}$;

---

## B  Additional details on informed search algorithms

In tables 3, 4, 5, 6, and 7 we present the hyperparameters used in our implementation for all baselines. Usage of each hyperparameter can be found in our codebase. We elaborate below on hyperparameters specific to our contribution, namely A$^*$-NEAR and IDS-BB-NEAR.

In A$^*$-NEAR and IDS-BB-NEAR, we allow for a number of hyperparameters to be used that can additionally speed up our search. To improve efficiency, we allow for the frontier in these searches to be bounded by a constant size. In doing so, we sacrifice the completeness guarantees discussed in the main text in exchange for additional efficiency. We also allow for a scalar performance multiplier, which is a number greater than zero, that is applied to each node in the frontier when a goal node is found. The nodes on the frontier must have a lower cost than the goal node after this performance multiplier is applied; otherwise, they are pruned from the frontier in the case of branch-and-bound. When considering non-goal nodes, this multiplier is not applied. We introduce an additional parameter that decreases this performance multiplier as nodes get farther from the root; i.e become more complete programs. We also decrease the number of units given to a neural network within a *neural program approximation* as nodes get further from the root, with the intuition that neural program induction done in a more complete program will likely have less complex behavior to induce. We also allow for the branching factor of all nodes in the graph to be bounded to a user-specified width in order to bound the combinatorial explosion of program space. This constraint comes at the expected sacrifice of completeness in our program search, given that potentially optimal paths are arbitrarily not considered.

In our experiments, we show that using these approximative hyperparameters allows for an acclerated search while maintaining strong empirical results with our NEAR-guided search algorithms.

|  | feature dim | label dim | max seq len | # train | # valid | # test |
|---|---|---|---|---|---|---|
| CRIM13-sniff | 19 | 2 | 100 | 12404 | 3007 | 2953 |
| CRIM13-other | 19 | 2 | 100 | 12404 | 3007 | 2953 |
| Fly-vs.-Fly | 53 | 7 | 300 | 5339 | 594 | 1048 |
| Bball-ballhandler | 22 | 6 | 25 | 18000 | 2801 | 2893 |

Table 2: Dataset details.

|  | max depth | init. # units | min # units | max # children | penalty | $\beta$ |
|---|---|---|---|---|---|---|
| CRIM13-sniff | 10 | 15 | 6 | 8 | 0.01 | 1.0 |
| CRIM13-other | 10 | 15 | 6 | 8 | 0.01 | 1.0 |
| Fly-vs.-Fly | 6 | 25 | 10 | 6 | 0.01 | 1.0 |
| Bball-ballhandler | 8 | 16 | 4 | 8 | 0.01 | 1.0 |

Table 3: Hyperparameters for constructing graph $\mathcal{G}$.

|  | # LSTM units | # epochs | learning rate | batch size |
|---|---|---|---|---|
| CRIM13-sniff | 100 | 50 | 0.001 | 50 |
| CRIM13-other | 100 | 50 | 0.001 | 50 |
| Fly-vs.-Fly | 80 | 40 | 0.00025 | 30 |
| Bball-ballhandler | 64 | 15 | 0.01 | 50 |

Table 4: Training hyperparameters for RNN baseline.

|  | # neural epochs | # symbolic epochs | learning rate | batch size |
|---|---|---|---|---|
| CRIM13-sniff | 6 | 15 | 0.001 | 50 |
| CRIM13-other | 6 | 15 | 0.001 | 50 |
| Fly-vs.-Fly | 6 | 25 | 0.00025 | 30 |
| Bball-ballhandler | 4 | 6 | 0.02 | 50 |

Table 5: Training hyperparameters for all program learning algorithms. The # neural epochs hyperparameter refers only to the number of epochs that neural program approximations were trained in NEAR strategies.

|  | A*-Near | IDS-bb-Near | | | |
|---|---|---|---|---|---|
|  | frontier size | frontier size | init. depth | depth bias | perf. mult. |
| CRIM13-sniff | 8 | 8 | 5 | 0.95 | 0.975 |
| CRIM13-other | 8 | 8 | 5 | 0.95 | 0.975 |
| Fly-vs.Fly | 10 | 10 | 4 | 0.9 | 0.95 |
| Bball-ballhander | 400 | 30 | 3 | 1.0 | 1.0 |

Table 6: Additional hyperparameters for A*-Near and IDS-bb-Near.

|  | MC(TS) | Enum. | Genetic | | | | | | |
|---|---|---|---|---|---|---|---|---|---|
|  | samples /step | max # prog. | pop. size | select. size | # gens | total # evals | mutate prob. | enum. depth |
| CRIM13-sniff | 50 | 300 | 15 | 8 | 20 | 100 | 0.1 | 5 |
| CRIM13-other | 50 | 300 | 15 | 8 | 20 | 100 | 0.1 | 5 |
| Fly-vs.Fly | 25 | 100 | 20 | 10 | 10 | 10 | 0.1 | 6 |
| Bball-ballhander | 150 | 1200 | 100 | 50 | 10 | 1000 | 0.01 | 7 |

Table 7: Additional hyperparameters for other program learning baselines

## C  Details of Experimental Domains

### C.1  Fly-v.-Fly

The *Fly-vs.-Fly* dataset [14] tracks a pair of flies and their actions as they interact in different contexts. Each timestep is represented by a 53-dimensional feature vector including 17 features outlining the fly's position and orientation along with 36 position-invariant features, such as linear and angular velocities. Our task in this domain is that of *bout-level classification*, where we are tasked to classify a given trajectory of timesteps to a corresponding single action taking place. Of the three datasets within *Fly-vs.-Fly*, we use the *Aggression* and *Boy-meets-Boy* datasets and classify trajectories over the 7 labeled actions displaying aggressive, threatening, and nonthreatening behaviors in these two datasets. We omit the use of the *Courtship* dataset for our classification task, primarily due to the heavily skewed trajectories in this dataset that vary highly in length and action type from the *Aggression* and *Boy-meets-Boy* datasets. Full details on these datasets, as well as where to download them, can be found in [14]. To ensure a desired balance in our training set, we limit the length of trajectories to 300 timesteps, and break up trajectories that exceed this length into separate trajectories with the same action label for data augmentation. Our training dataset has 5339 trajectories, our validation set has 594 trajectories, and our test set has 1048 trajectories. The average length of a trajectory is 42.06 timesteps.

**Training details of Fly-v.-Fly baselines.** For all of our program synthesis baselines , we used the Adam [22] optimizer and cross-entropy loss. Each synthesis baseline was run on an Intel 4.9-GHz i7 CPU with 8 cores, equipped with an NVIDIA RTX 2070 GPU w/ 2304 CUDA cores.

### C.2  CRIM13

The CRIM13 dataset studies the social behavior of a pair of mice annotated each frame by behavior experts [5] at 25Hz. The interaction between a resident mouse and an intruder mouse, which is introduced to the cage of the resident, is recorded. Each mice is tracked by one keypoint and a 19 dimensional feature vector based on this tracking data is provided at each frame. The feature vector consists of features such as velocity, acceleration, distance between mice, angle and angle changes. Our task in this domain is *sequence classification*: we classify each frame with a behavior label from CRIM13. Every frame is labelled with one of 12 actions, or "other". The "other" class corresponds to cases where no action of interest is occurring. Here, we focus on two binary classification tasks: other vs. rest, and sniff vs. rest. The first task, other vs. rest, corresponds to labeling whether there is

| | CRIM13-sniff | | | CRIM13-other | | | Fly-vs.-Fly | | | Bball-ballhandler | | |
|---|---|---|---|---|---|---|---|---|---|---|---|---|
| | Acc. | F1 | Depth | Acc. | F1 | Depth | Acc. | F1 | Depth | Acc. | F1 | Depth |
| Enum. | .024 | .105 | 1 | .036 | .011 | 1 | .013 | .012 | 0 | .009 | .009 | 0.6 |
| MC | .013 | .127 | 1.7 | .088 | .031 | 0.6 | .028 | .018 | 2 | .012 | .012 | 0.6 |
| MCTS | .047 | .076 | 0 | .103 | .036 | 0 | .008 | .009 | 0.94 | .003 | .002 | 0 |
| Genetic | .003 | .015 | 0.6 | .005 | .004 | 1.7 | .028 | .030 | 1 | .016 | .019 | 0.6 |
| IDDFS-NEAR | .021 | .056 | 2 | .006 | .005 | 0.6 | .023 | .016 | 0 | .006 | .006 | 0 |
| A*-NEAR | .026 | .114 | 1.7 | .030 | .010 | 2.1 | .003 | .004 | 0 | .034 | .034 | 0 |
| RNN | .008 | .019 | - | .005 | .002 | - | .006 | .005 | - | .001 | .001 | - |

Table 8: Standard Deviations of accuracy, F1-score, and program depth of learned programs (3 trials).

an action of interest in the frame. The second task, sniff vs. rest, corresponds to whether the resident mouse is sniffing any part of the intruder mouse. These two tasks are chosen such that the RNN baseline has reasonable performance only using the tracked keypoint features of the mice. We split the train set in [5] at the video level into our train and validation set, and we present test set results on the same set as [5]. Each video is split into sequences of 100 frames. There are 12404 training trajectories, 3077 validation trajectories, and 2953 test trajectories.

We observed higher variance in F1 score for the CRIM13-sniff class in Table 8, as compared to the other experiments. For this particular class, due to the high variance of both baseline and NEAR runs, we would like to note the importance of repeating runs.

**Training details of CRIM13 baselines.** All CRIM13 baselines training uses the Adam [22] optimizer and cross-entropy loss. In the loss for sniff vs. rest, the sniff class is weighted by 1.5. Each synthesis baseline was run on an Intel 2.2-GHz Xeon CPU with 4 cores, equipped with an NVIDIA Tesla P100 GPU with 3584 CUDA cores.

## C.3 Basketball

The basketball data tracks player positions ($xy$-coordinates on court) from real professional games. We used the processed version from [50], which includes trajectories over 8 seconds (3 Hz in our case of sequence length 25) centered on the left half-court. Among the offensive and defensive teams, players are ordered based on their relative positions. Labels for the ballhandler were extracted with a labeling function written by a domain expert. See Table 2 for full details of this dataset.

**Training details of Basketball baselines.** All Basketball experiments use Adam [22] and optimize cross-entropy loss. Each synthesis baseline was run on an Intel 3.6-GHz i7-7700 CPU with 4 cores, equipped with an NVIDIA GTX 1080 Ti GPU with 3584 CUDA cores.