[Reviews · NeurIPS 2020]

Review 1

Summary and Contributions: In this paper, the authors propose using admissible neural heuristic in classic informed search algorithms for searching the structure of differentiable programs. During the program search, the unrealized part of a program is replaced with a neural network to estimate the cost-to-go value in that direction. Empirical results demonstrate the proposed algorithm is able to find interpretable programs with lower errors than other search methods in the given time limit.

Strengths: + Idea is clear and novel. Using neural networks to estimate the model fitting performance upperbound seems to be a straight-forward but effective choice. It's very interesting to see classic AI search algorithms can be combined with deep learning in such a way. + Convincing experiment results and analysis. + Possible connections to multiple research fields in machine learning, including learning interpretable policies in RL, learning modular networks in VQA etc. + Excellent paper writing, very logical and easy to understand.

Weaknesses: - The benchmark datasets are relatively toyish. But this is understandable since the focus here is to learn interpretable programs.

Correctness: yes

Clarity: yes

Relation to Prior Work: yes

Reproducibility: Yes

Additional Feedback: Suggestions: 1. The authors need to include MCTS as one of the baseline to make it more comprehensive, although MCTS might take a long time to run. 2. Consider adding a synthetic experiment where there are ground truth programs for reference. Since A* is guaranteed to find the optimal solution, it would be interesting to see if the algorithm can recover the ground truth program. The authors can also empirically verify the epsilon-admissibility of the neural net heuristics. 3. Include more realistic experiments such as inducing a program (modular network) in Visual QA problems where only the images and the answers is given. Edit: I've read the response and decided to keep my score.


Review 2

Summary and Contributions: This work considers the problem of synthesizing programs from input/outputs, but where some of the components of the program might have continuous parameters, and where the entire program is differentiable with respect to these parameters. Neurosymbolic programs are a special case of this set up (symbolic programs which can call out to neural modules if needed). This is an especially challenging combinatorial search problem, because not only do we have to consider an infinitely large, discrete space of program structures, but we also have to consider an inner-loop optimization over continuous parameters. The approach they take is to perform an explicit symbolic graph search over the discrete space of partial programs. As a heuristic function for this graph search, they train neural networks to approximate the behavior of incomplete portions of the program syntax tree. Because certain neural networks are universal function approximators, they reason that the performance of this neural-network-plus-partial-program is an upper bound on the performance of any prolongation of that partial program. To the extent that this assumption holds, the resulting heuristic is admissible in the A* sense. They show this approach outperforms a number of reasonable program synthesis baselines on some classification problems, but (imo) importantly is that the model is able to recover differentiable programs which rival the performance of blackbox neural networks yet are easily interpretable.

Strengths: The technical approach is intuitive yet nonobvious and appears novel. Most importantly: the work is extremely relevant to this conference, relevant to both interpretability and neurosymbolic integration, and the solution they propose is elegant. I can see their approach being useful and influential in the future. It is also theoretically well motivated, building on classic work in heuristic search. They compare with exactly the program synthesis baselines I would like to see them compare with.

Weaknesses: The particular data sets that they apply this work to seem nonstandard/unusual. Why did you choose these data sets? The interpretability angle seems underexplored relative to its promise, but I think this is acceptable given the page limits. If this paper is accepted you will have an extra page and I recommend using that extra page to dwell longer on interpretability - for example, can you show some of the programs illustrated in the trade-off of Figure 6?

Correctness: The claims and methods seem correct.

Clarity: The manuscript is clear and easy to read -- no doubt in large part because the method is intuitive and elegant, but the writing is solid too.

Relation to Prior Work: The related works section is comprehensive. However, I'm not intimately familiar with the prior literature on "Structure search using relaxations" outside of classic applications like integer linear programming and modern incarnations such as DARTS. If another reviewer could comment on the section of related work labeled "Structure search using relaxations" it would be appreciated.

Reproducibility: Yes

Additional Feedback: If you want a model that is both accurate and interpretable, a simple thing you could do is to learn a small neural network (or even a logistic regressor/etc.) that predicts the residual after running your interpretable model. Did you consider this? What are the prospects of combining this with richer neural modules, such as those in the HOUDINI system (Valkov et al 2018), or architectures such as modular meta-learning (Alet et al 2018)? I think you either don't need to show algorithm 1 (which is literally just vanilla A*), or you need to expand it to be specialized to your setting. ----- Post rebuttal: Based on input from the other reviewers: It might be a good idea to compare with terpret-style approaches, which would require finitizing the program space by unrolling the CFG over programs (tbl 1 suggests it would need to be unrolled to a depth of ~8). There is the issue of which terpret backend to use: SAT won't do because of the continuous parameters. You could then solve for the minimum cost program in that finite space using either (1) an SMT solver, or (2) gradient descent. At a minimum, the text should cite terpret, and point out that this work differs by not restricting itself to finite spaces of programs, and also by scaling to fairly deep (~8) mixed discrete/continuous programs.


Review 3

Summary and Contributions: In this paper, the authors propose an approach with admissible neural heuristics to learn differentiable programs, in which the authors frame this attractive research problem as a relaxation search issue in a graph, and use three different kinds of datasets to evaluate the performance of proposed algorithms in comparison with three baseline models. The contributions are summarized as followed: 1. A novel approach based on optimization relaxation to learn a differential program from a set of training datasets. 2. A formal program formulation based on programming language definition is clear to express the proposed idea. 3. Integrating Near with heuristic graph search algorithms and Neural Relaxations as Admissible Heuristics can limit search space and improve whole system performance.

Strengths: See the above contributions.

Weaknesses: 1. Figure 1 gives a set of DSL grammar definition, however the meaning is unclear. For example, which one is nonterminal or terminal symbol? In addition, there is no detailed description about basic algebraic operations and parameterized library functions. 2. The authors proposed to use various classes of neural networks as relaxations of partial programs; however, there is no content about this statement. It would be better to list some specific neural networks. 3. In Section 2, the authors declared that there is a simple type system to ensure type consistent. However, I cannot find any details about this topic in the paper. 4. In the experiments, even the authors use three different datasets to evaluate performance of proposed approach, they depict a similar scenario about how trajectory would change over a specific set of actions. I doubt about whether these similar datasets can validate performance improvement of NEAR. In addition, there is no clear description to explain why the RNN-based approach has a better performance than the baseline models and A*-NEAR in Table 1.

Correctness: Based on the experiments and author’s presentation, the proposed approach has some innovations, although the benchmarks chosen are not thorough.

Clarity: The paper is well written but should be careful with abbreviation usage since some abbreviations are defined multiple times, for example “Neural Admissible Relaxation”.

Relation to Prior Work: I did not find any big poblems about related work about three different research domain: Neural Program Induction, DSL-based Program Synthesis and Structure Search using Relaxations.

Reproducibility: Yes

Additional Feedback:


Review 4

Summary and Contributions: This paper provides a method for learning differentiable programs through a top-down search where a each partial program is replaced with an neural network surrogate that can be considered a heuristic for the resulting program. Because the underlying DSL can only represent differentiable programs this provides an admissible heuristic for program synthesis that can be then inserted into some informed search for programs.

Strengths: The approach proposed in the paper seems sound and reasonable. As search-based methods seem to be a major way we are learning complex programs in the literature, additional approaches in this spirit can really contribute to work being done in the community.

Weaknesses: This work doesn't seem very novel. Differentiable programs have been explored in the work of TrepeT (https://arxiv.org/abs/1608.04428, https://arxiv.org/abs/1611.02109) and ultimately seems to be yet another heuristic for performing a search over program space. The evaluation is only against baseline methods and not even some of the related work the authors mention themselves. The work would greatly benefit from being compared against something other than baselines and some tests on generalisation The work is only evaluated compared to baselines and not even compared to the related work mentioned in the paper.

Correctness: The work appears to be correct and applicable to the problem. The methodology seems sound and reasonable if a bit incomplete.

Clarity: This work is not very clearly written. It was challenging to understand the main contributions as they are only discussed in the introduction and never mentioned again. Much of the details I wanted about NEAR such as the kinds of neural architectures used and how exactly these surrogates were trained is not mentioned in the main paper or the appendix except in passing. Update with rebuttal: The author response and subsequent discussion really clarified the contribution of the work and if some of that can make its way into the paper it will be stronger for it.

Relation to Prior Work: Prior work is discussed but some relevant literature is missed but more importantly the work is not compared in detail to even some of the prior literature discussed. It would have been nice to see it compared to use neural search-based synthesis approaches even if the other approaches were more general and less fit for the given problems. Update with rebuttal: I agree with the authors that it can be challenging to compare to existing work by say restricting the program space to something both NEAR and say Trepet (does not need to be Trepet) for example can both represent. I am hopeful that NEAR will behave favourably but I need to see it.

Reproducibility: Yes

Additional Feedback:

[Author Response · NeurIPS 2020]

Thank you for your constructive feedback. We address the reviewers' main points below; however, we will also incorporate all other feedback in the reviews into the paper's final version.

**The motivation is unclear (R3, R4), in particular, as differentiable programming is not new (R4).** While differentiable programming is indeed not new, in virtually all prior efforts on the topic, including the TerpreT approach that R4 mentions, one considers a *parameterized* representation of programs (essentially, an "architecture" in the language of our paper), and the learning objective is to find optimal parameters for this representation. In contrast, our approach searches over the nonparameteric space of all architectures expressed in a rich programming language. The only paper we know that searches over architectures of differentiable programs is "Houdini: Lifelong Learning as Program Synthesis" (Valkov et al., NeurIPS '18). There, architecture search is performed using type-guided enumeration and an evolutionary algorithm, i.e., two of the baselines that we beat.

The problem of searching in spaces of program architectures is also well-studied in classical program synthesis. The best-performing approaches there are based on enumeration, Monte Carlo sampling, and evolutionary algorithms (i.e., our baselines). Our paper's new observation, as noted by R1 and R2, is that we can do better than these approaches when programs are differentiable, by learning approximately admissible search heuristics and using these heuristics to guide an informed search. A version of this idea has previously come up in LASSO search (see the Related Work section); however, the idea is completely new to the program learning and differentiable programming literatures.

**We do not experimentally compare against the approaches in the related work section (R4)**: We compare with all approaches in the related work section with which a meaningful comparison is possible. Specifically, we cannot compare against neural program induction techniques, as the outputs of these approaches are neural nets as opposed to programs. DARTS-style, gradient-based architecture search cannot be naturally extended to our setting because of the complexity of our programming language (see discussion in lines 294-299). We cannot compare against metalearning-based approaches such as "Accelerating Search-Based Program Synthesis Using Learned Probabilistic Models" (Lee et al., PLDI '18) and "DeepCoder: Learning to Write Programs" (Balog et al., ICLR '17) as well as RL-based approaches, because we do not have available a corpus of datasets and corresponding programs and must learn a program from a single dataset. We have already compared against enumerative and evolutionary architecture search, used by Valkov et al. We have also compared against Monte Carlo sampling, which underlies many of Bayesian program synthesis approaches, such as in "Sampling for Bayesian Program Synthesis" (Ellis et al., NeurIPS '16). We will be happy to compare with a MCTS baseline, as R1 recommends.

**The datasets used are nonstandard/toyish (R1, R2).** As R1 points out, a central focus of our work was on generating *interpretable* programs; to that end, we focused on behavior analysis applications where interpretability is an especially important concern. Prior efforts in machine learning research have used these datasets — see "Generating Multi-Agent Trajectories using Programmatic Weak Supervision" (Zhan et al., ICLR '19), "Learning Recurrent Representations for Hierarchical Behavior Modeling" (Eyjolfsdottir et al., ICML '18), "Learning fine-grained spatial models for dynamic sports play prediction" (Yue et al., ICDM '14), "Social behavior recognition in continuous video" (Burgos-Artizzu et al., CVPR '12). Also, these datasets are representative of real behavior data used by real domain scientists (sports analysts for basketball, and neuroscientists for CRIM13 and Fly-vs-Fly), and were used in real domain applications before use in machine learning research.

**Clarifications on the DSL (R3).** We will add further clarifying details about the DSL in the final version. We elided details of the algebraic operations and parameterized functions in the paper because these details do not affect the abstract form of our programs or the way our search algorithm works. The abstract notation for the language that we use in the paper is standard in research on Programming Languages (PL), and also appears in many prior papers in the intersection of PL and machine learning. As for our type system, it is quite basic and primarily ensures that the types of formal parameters, actual parameters, and return values are as expected; when expanding a partial architecture, we ensure that the chosen expansion is consistent w.r.t. this type system. Again, we will be happy to provide additional details in the final version if the paper is accepted.

**Questions regarding the neural models used in our heuristic (R3, R4).** As briefly mentioned in the paper and appendix, we use feedforward neural networks and LSTM recurrent neural networks as the neural architectures for our heuristic. The parameterization of these models varies depending on the complexity of the program using these models as relaxations. We provide the specific implementation of this in our codebase, but we will elaborate on this point in the appendix.

**Other Comments.** In addition to these main points, we want to thank the reviewers for their helpful suggestions on how to further improve the paper. We plan on incorporating and discussing all of these, such as a greater focus on interpretability (R2), incorporation of an MCTS baseline (R1), usage of a smaller synthetic experiment (R1), and the learning of a residual network post-program generation (R2). The detailed discussion on further clarifications or interesting future directions are deeply appreciated as well.

[Meta-Review · NeurIPS 2020]

Generating programs has been a long-standing problem in AI for many decades. Reviewers found valuable the fact that this approach combines prior literature on heuristic search with a modern neural networks approach to improve performance. Reviewers also found that methods which combine discrete and continuous parts of programs are in short supply, making this of wide interest and likely to spur further research. The fact that the approach is in a sense straightforward conceptually but not obvious while being able to perform when more complex methods like TerpreT are not suitable was also pointed out as a significant advance. Reviewers wished to see more related to the interpretability of the acquired models. Just because a model is not a neural network, one cannot say it is interpretable, for example a large decision tree is no more interpretable than a deep network. Evaluating this aspect would significantly strengthen the manuscript. Reviewers were concerned with the parameterization of the LSTMs, the fact that hyperparameters must be tuned to match the complexity of the expected programs for each dataset. Reviewers also wished to see more information on the type system used. Reviewers pointed out that the paper would be strengthened by a comparison against MCTS, and in particular against TerpreT, even if that latter comparison will necessarily be limited, and provided ideas on how to do so. Even the lowest-ranking reviewer agreed that with such an experiment they would be happy to accept, and even the highest-ranking reviewer agreed that such an experiment would significantly improve the paper. Differences in the review scores boiled down essentially to what one might expect these results to show and how comfortable one is to publish without them, with three out of four reviewers being comfortable doing so. We highly encourage the authors to include these experiments as they strengthen the already significant conceptual contribution.